# Diversity, Plausibility, and Difficulty: Dynamic Data-Free Quantization

## Abstract

Without access to the original training data, data-free quantization (DFQ) aims to recover the performance loss induced by quantization. Most previous works have focused on using an original network to extract the train data information, which is instilled into surrogate synthesized images. However, existing DFQ methods do not take into account important aspects of quantization: the extent of a computational-cost-and-accuracy trade-off varies for each image depending on its task difficulty. Neglecting such aspects, previous works have resorted to the same-bit-width quantization. By contrast, without the original training data, we make dynamic quantization possible by modeling varying extents of task difficulties in synthesized data. To do so, we first note that networks are often confused with similar classes. Thus, we generate plausibly difficult images with soft labels, where the probabilities are allocated to a group of similar classes. Under data-free setting, we show that the class similarity information can be obtained from the similarities of corresponding weights in the classification layer. Using the class similarity, we generate plausible images of diverse difficulty levels, which enable us to train our framework to dynamically handle the varying trade-off. Consequently, we demonstrate that our first dynamic data-free quantization pipeline, dubbed DynaDFQ, achieves a better accuracy-complexity trade-off than existing data-free quantization approaches across various settings.

## 1 Introduction

Despite the breakthrough advances in deep convolutional neural networks, their practical deployment remains limited due to the heavy computation and memory consumption. Such critical limitation has led to a burst of interests in various methodologies to improve efficiency, one of them being network quantization, which aims to reduce the bit-widths of weights or activation values. Among diverse lines of research in network quantization, quantization-aware training successfully obtains efficient networks without sacrificing accuracy by fine-tuning a network after quantization with the original training data (Zhou et al., 2016; Jacob et al., 2018; Jung et al., 2019; Esser et al., 2020).

However, due to data privacy and security concerns, the original training data are often inaccessible in real-world applications such as the medical, military, and industrial domains (Sun et al., 2017; Kanazawa et al., 2019). This raises a demand for quantization *without* the original training dataset, referred to as data-free quantization (DFQ). Recent works on DFQ aim to generate useful pseudo data by extracting and utilizing the knowledge of the train data concealed in the original pre-trained network, such as running statistics stored in the batch normalization layers (Cai et al., 2020; Shoukai et al., 2020; Zhang et al., 2021; Choi et al., 2021). Then, they use the synthesized data to update the parameters of the quantized network.

Yet, existing methods overlook a crucial aspect of quantization: the trade-off between computational cost and accuracy varies for each image, as shown in Figure 1. Despite the fact that adopting different bit-width for images of different classification difficulties allows a better trade-off (Liu et al., 2022; Hong et al., 2022; Tian et al., 2023), existing methods adopt the same bit-width for all images. Nevertheless, the realization of such image-wise bit-width allocation is challenging without the assistance of the original training data. To allocate appropriate bit-widths for natural test images, which are largely diverse in terms of classification difficulty, the network should be trained with difficulty-varying samples (ranging from easy to difficult) beforehand. Unfortunately,

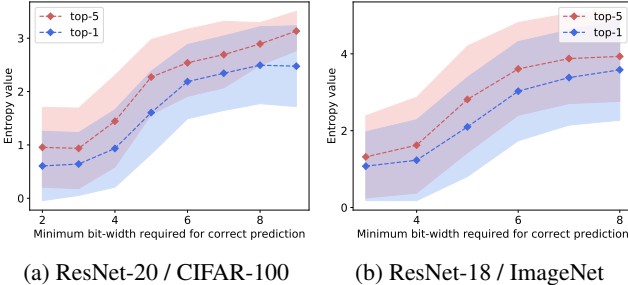

(a) ResNet-20 / CIFAR-100     (b) ResNet-18 / ImageNet

Figure 1: **Correlation between quantization sensitivity (x-axis) and prediction entropy (y-axis) of an image**. An image whose output prediction is low in entropy can be inferenced with a lower-bit model without harming the prediction accuracy. This motivates us to leverage high (low) bit-widths for samples of high (low) entropy for a better trade-off of accuracy and efficiency.

the existing works on DFQ fail to generate data of diverse difficulties (Cai et al., 2020; Shoukai et al., 2020; Zhang et al., 2021; Zhong et al., 2022). Consequently, this leads to a failure in dynamic quantization, where computational resources are not effectively distributed across the test images.

On this basis, we propose the first **D**ata-**F**ree **Dyna**mic **Q**uantization framework (DynaDFQ) that learns to allocate bit-widths dynamically to the input image according to its difficulty, without the need of the original training data. To this end, we claim that two aspects of difficulty should be considered: diversity and plausibility. As discussed above, the difficulty diversity (ranging from easy to difficult) is needed for the network to learn how to allocate computational resources according to the difficulty. Furthermore, the plausibility of difficulty (e.g., car vs. truck) needs to be modeled such that the learned quantization scheme can be generalized to real images. To formulate both aspects in our image synthesis, we make use of readily available information that has not been exploited by previous works: class similarity. Without access to the original dataset, we show that class similarity information can be extracted from the classification layer of the original pre-trained network (*i.e.*, the more similar the weights are, the more similar corresponding classes are), as illustrated in Figure 2. Using the class similarity, we can achieve both difficulty diversity and plausibility by synthesizing images whose soft label weights are randomly allocated among similar classes.

Then, the generated images and soft labels are used to train our dynamic quantization network that adopts different layer-wise bit-widths for each image. Simply minimizing original cross-entropy loss functions will encourage the network to maximize accuracy, choosing the highest bit-width. To better handle the trade-off, we use a bit regularization loss that penalizes the computational resources (bit-FLOPs) of the dynamically quantized network. Among our generated samples of diverse difficulties (ranging from easy to difficult samples), easy samples will be learned very quickly, leading to very small loss values and giving almost no contribution to the training afterward. Thus, we apply mixup (Zhang et al., 2017) on easy samples to further expose the network to more diverse levels of difficulties. The experimental results demonstrate the outstanding performance of DynaDFQ across various image classification networks, underlining the effectiveness of our difficulty-diverse sample generation in bridging the gap between dynamic quantization and data-free quantization.

## 2 RELATED WORKS

**Data-free quantization (DFQ)** Network quantization has led to a dramatic reduction in computational resources without much compromise of network accuracy (Zhou et al., 2016; Jung et al., 2019; Rastegari et al., 2016; Choi et al., 2018; Courbariaux et al., 2015; Banner et al., 2019; Krishnamoorthi, 2018). However, the arising concerns regarding data privacy and security have led to the lack of the access to the original training data (Sun et al., 2017; Kanazawa et al., 2019), which motivated the community to investigate network quantization methods that do not require the original training data, namely, DFQ. Nagel et al. (2019) first propose to directly adapt the weights of the pre-trained model to reduce the quantization error. Subsequent works fine-tune the quantized network with synthetic data generated via random sampling (Cai et al., 2020) or from generative models (Shoukai et al., 2020; Liu et al., 2021) to match the statistics of the pre-trained network. The recent focus of DFQ methods is to generate synthetic samples that better match the real data. In specific, Zhang

et al. (2021) aim to better mimic the feature statistics of the real data, Zhong et al. (2022) preserve the intra-class heterogeneity, and He et al. (2021) apply ensemble technique to several compressed models to produce hard samples. More recently, Choi et al. (2021) synthesize additional boundary-supporting samples, and Li et al. (2023) focus the generation process on hard-to-fit samples, which are samples with low predicted probability on ground truth labels. However, these methods either neglect that the real data samples exhibit greatly different classification *difficulties* or neglect the semantic relationship between classes for difficult samples. In contrast, we generate samples of different difficulties with plausibility by considering the similarity between classes, which we find to be beneficial for dynamically allocating bit-widths for various samples.

**Dynamic inference** Different input images encounter different task difficulties (*i.e.*, they have different required computational resources for processing). Accordingly, the dynamic computational resource adaptation for different input images has been widely studied in computer vision tasks via controlling the quantization bit-width (Liu et al., 2022; Hong et al., 2022; Tian et al., 2023), or number of channels (Wang et al., 2021; Oh et al., 2022), convolutional layers (Wu et al., 2018; Liu et al., 2020; Kong et al., 2021; Yu et al., 2021; Xie et al., 2021). Dynamic quantization has been recently investigated, as dynamically controlling the bit-widths grants a better trade-off between accuracy and computational complexity. Specifically, Liu et al. (2022) effectively allocates bit-widths to different inputs and layers on image classification task. Dynamic bit allocation has also been proven effective on other tasks, such as video recognition (Sun et al., 2021) and image restoration (Hong et al., 2022; Tian et al., 2023). However, these approaches require the training of dynamic networks with the original or even additional training dataset. Unfortunately, these datasets are not always available in real-world applications. To cope with such issues, we, for the first time, present a practical solution to achieve dynamic quantization without access to the original training dataset.

## 3 PROPOSED METHOD

In the following sections, we introduce the general procedure for data-free quantization (Sec. 3.1) and elaborate on our DynaDFQ. We observe that the classification difficulty of an image, which we estimate with prediction entropy, can hint the optimal bit-width for processing each image. Motivated by the observation, we first generate fake images with various classification difficulties (Sec. 3.2). Then, to achieve a better trade-off between computational resources and accuracy, we employ a dynamic quantization network that assigns higher bit-widths to images in need (Sec. 3.3). Moreover, our training scheme allows for a better trade-off with respect to each image (Sec. 3.4).

### 3.1 PRELIMINARIES

Data-free quantization aims to obtain an accurate quantized network $\mathcal{Q}$ from a pre-trained floating-point (FP) 32-bit network $\mathcal{P}$ without the assistance of the original training data. DFQ approaches (Cai et al., 2020; Zhong et al., 2022) commonly adopt a two-stage approach: (1) synthetic data generation and (2) quantization-aware training with the synthetic data. First, synthetic data pairs $\{I_i, y_i\}_{i=1}^N$ are initialized by randomly initializing a synthetic image $I_i$ with Gaussian noise and its label $y_i$ as arbitrary class (*i.e.*, one-hot label). Then, synthetic images are generating by optimizing the following objective:

$$\mathcal{L}^G = \frac{1}{N} \sum_{i=1}^N \mathcal{L}_{ce}(\mathcal{P}(I_i), y_i) + \frac{1}{N} \sum_{i=1}^N \mathcal{L}_{bns}(I_i), \tag{1}$$

where the first term $\mathcal{L}_{ce}$ is the cross-entropy loss that guides each $I_i$ to follow its label $y_i$ w.r.t. $\mathcal{P}$. The second term $\mathcal{L}_{bns}$ encourages the intermediate statistics (mean and standard deviation) produced by $\mathcal{P}$ to match the statistics stored in the batch normalization (BN) layers of $\mathcal{P}$.

Subsequently, the obtained synthetic samples are used to fine-tune the quantized network $\mathcal{Q}$. $\mathcal{Q}$ is first derived by quantizing weights and features of $\mathcal{P}$. The input feature $X$ of each convolutional layer is quantized to $X_q$ with bit-width $b$ by

$$X_q \equiv q_b(X) = \lfloor \frac{\text{clamp}(X, l, u) - l}{s(b)} \rceil \cdot s(b) + l, \tag{2}$$

where $s(b) = (u - l)/(2^b - 1)$, $l$ and $u$ are learnable scale parameters. First, clamp($\cdot, l, u$) truncates $X$ into the range of $[l, u]$ and then scaled to $[0, 2^b - 1]$. The features are then rounded to the nearest

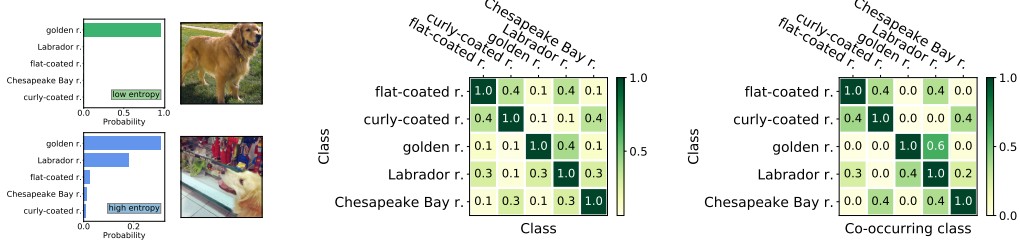

(a) Easy/difficult sample     (b) Classification weight similarity     (c) Class co-occurrences in prediction

Figure 2: **Motivation of class similarity-based generation.** (a) While the prediction output of an easy sample is nearly one-hot, that of a confusing sample is a soft distribution over similar classes. (b-c) Similar classes can be obtained without access to the original data by measuring the similarity between pre-trained classification weights. The measured similarity is correlated to the class co-occurrences in the prediction of the original data. We visualize five retriever classes of ImageNet.

integer with $\lfloor \cdot \rceil$, and the integers are re-scaled back to range $[l, u]$. Following Zhong et al. (2022); Li et al. (2023), we use a layer-wise quantization function for features and a channel-wise quantization function for weights. The quantization parameters $l$, $u$, and the weights of $\mathcal{Q}$ are trained by the optimization as follows:

$$\mathcal{L}^Q = \frac{1}{N} \sum_{i=1}^{N} \mathcal{L}_{ce}(\mathcal{Q}(\boldsymbol{I}_i), \boldsymbol{y}_i) + \frac{1}{N} \sum_{i=1}^{N} \mathcal{L}_{kd}(\mathcal{P}(\boldsymbol{I}_i), \mathcal{Q}(\boldsymbol{I}_i)), \tag{3}$$

where $\mathcal{L}_{kd}$ is Kullback-Leibler divergence loss.

## 3.2 SIMILARLITY-BASED PLAUSIBLE-AND-DIVERSE-DIFFICULTY DATA GENERATION

**Difficulty diversity** It is acknowledged that different images have different quantization sensitivities Liu et al. (2022), which can be defined as the degree of quantization at which the correct prediction can remain intact. Samples whose predictions remain accurate even after intense (low-bit) quantization are considered less sensitive to quantization. Intuitively, allocating higher bit-widths to quantization-sensitive samples can lead to finding a better trade-off between efficiency and accuracy. However, directly measuring the quantization sensitivity is practically infeasible, as it requires multiple inferences of each sample with different bit-width networks that are separately trained.

Instead, we find that the quantization sensitivity is related to the classification difficulty of samples. The classification difficulty can be modeled with entropy (Dehghan & Ghassemian, 2006), which can be used to estimate the uncertainty of the model prediction. We formulate the entropy of an sample $\boldsymbol{I}_i$ as the entropy of the prediction using $\mathcal{P}$, such that:

$$H(\mathcal{P}(\boldsymbol{I}_i)) \equiv H(\boldsymbol{y}_i^{\mathcal{P}}) = \sum_{k=1}^{C} y_{i,k}^{\mathcal{P}} \log(y_{i,k}^{\mathcal{P}}), \tag{4}$$

where $y_{i,k}^{\mathcal{P}}$ denotes the output probability of $k$-th class from $\mathcal{P}(\boldsymbol{I}_i)$. As shown in Figure 1, we observe that the quantization sensitivity has a strong correlation with the entropy of the model prediction. Upon the observation, we estimate the quantization sensitivity by means of entropy. To this end, in order to effectively allocate bit-widths to images, we are motivated to generate pseudo images with diverse levels of difficulties. One simple approach to diversify the difficulty in the generated pseudo data is to assign a randomly sampled soft label to an image instead of an arbitrary class $y$. However, simple random sampling does not provide a control over the diversity of difficulty. Furthermore, it does not take the realistic difficulty into account, unable to guide the dynamic quantization framework to effectively allocate the computational costs for new real images.

**Difficulty Plausibility** To generate data that facilitate the training of dynamic quantization framework, it is important to generate images whose difficulties are plausible (i.e., similar to real images). According to our observation in Figure 2a, the prediction of images in the original data reflects the relations between classes; a difficult sample for the classification network is confusing between similar classes. Thus, our goal is to assign the images with soft labels that consider the similarity

between classes. To obtain the similarity between classes without access to the original data, we exploit the classification layer of the original pre-trained network, as each class weight vector is regarded as a useful estimated prototype of the class (Saito et al., 2019). To validate our assumption that class weight similarity can be used to measure the similarity between classes of the original data, we compare the similarity of two class weight vectors and the actual co-occurrences of the two classes in the prediction output of the original data. As visualized in Figure 2, the similarity matrix of class weights and class co-occurrences are closely correlated. Thus, we formulate the similarity between $j$-th class and $k$-th class as follows:

$$sim(j, k) = \boldsymbol{W}_j \times \boldsymbol{W}_k^T, \qquad j, k = 1, ..., C, \tag{5}$$

where $\boldsymbol{W}$ denotes the weight vectors of the classification layer and $\boldsymbol{W}_j$ denotes the weight vector that corresponds to the $j$-th class. Motivated by the previous observations, we assign the probability scores of soft labels only to top-$K$ similar classes, obtained using the similarity derived in Eq. 5. Given an arbitrarily sampled class $y$ for the $i$-th synthetic image $\boldsymbol{I}_i$, the soft label $\boldsymbol{y}_i \in \mathbb{R}^C$ is formulated as follows:

$$y_{i,k} = \begin{cases} z_k & \text{if } sim(y, k) \in \text{top-}K(\{sim(y, 1), ..., sim(y, C)\}), \\ 0 & \text{else}, \end{cases} \tag{6}$$

where $y_{i,k}$ is the $k$-th probability value of $\boldsymbol{y}_i$ and the probability score $\boldsymbol{z} = [z_1 \, z_2 \, \cdots \, z_K]^T$ are randomly sampled from Dirichlet distribution $\boldsymbol{z} \sim \text{Dir}(K, \boldsymbol{\alpha})$, in which simply set $\boldsymbol{\alpha}$ as $\boldsymbol{1}$. The difficulty of each sample can be controlled by $K$ and the entropy of the assigned soft label. For instance, if an image is assigned with a soft label with high entropy and large $K$, the image will likely be optimized to be confusing between many similar classes, making it a difficult sample. To generate images of diverse difficulty, we randomly sample the soft label for similar classes, and we generate $r$ ratio of samples optimized with top-$K$ similar classes and $1-r$ ratio of samples with top-1 class, where diversifying ratio $r$ is a hyper-parameter. Overall, the synthetic images are optimized with the objective function as follows:

$$\mathcal{L}^G = \frac{1}{N} \sum_{i=1}^{N} \mathcal{L}_{bns}(\boldsymbol{I}_i) + \beta \frac{1}{N} \sum_{i=1}^{N} \mathcal{L}_{ce}(\mathcal{P}(\boldsymbol{I}_i), \boldsymbol{y}_i), \tag{7}$$

where $\beta$ balances the losses. Given *diverse*, *plausibly* difficult and easy images, we can now facilitate the training of dynamic quantization framework.

### 3.3 DYNAMIC QUANTIZATION

Given the synthetic images and labels, we train a bit selector to allocate bit-width for each layer and image. Specifically, the bit selector learns to assign the image-wise probability for quantization functions of different bit-width candidates $\{b_m\}_{m=1}^{M}$ for each layer during training, where we set $M = 3$ in this work. To minimize the additional overhead, we use a lightweight bit selector that consists of a two-layer MLP followed by a Softmax operation, following Liu et al. (2022):

$$p_{b_m}(\boldsymbol{X}) = \text{Softmax}(\text{MLP}(\boldsymbol{X})), \tag{8}$$

where MLP consists of an average pooling layer followed by two fc layers with a dropout layer in between. During inference, for each input image and layer, a quantization function of bit-width with the highest probability is selected. Since selecting the max probability is a non-differentiable operation, we use a straight-through estimator (Bengio et al., 2013) to make the process differentiable:

$$\boldsymbol{X}_q = \begin{cases} \text{argmax}_{q_{b_m}(\boldsymbol{X})} p_{b_m} & \text{forward}, \\ \sum_m^M q_{b_m}(\boldsymbol{X}) \cdot p_{b_m}(\boldsymbol{X}) & \text{backward}. \end{cases} \tag{9}$$

Also, as dynamic quantization for weights requires storing the weight of the largest bit candidate, which occupies a large storage, we apply a bit selector to choose bit-widths for activations only.

### 3.4 QUANTIZATION-AWARE TRAINING

**Bit regularization loss** We train our dynamically quantized network $\mathcal{Q}$ and the bit selector using the synthesized difficulty-diverse training pairs $\{\boldsymbol{I}_i, \boldsymbol{y}_i\}_{i=1}^{N}$. Solely minimizing the original cross-entropy loss function can lead the bit selector to choose the highest bit-width to maximize the accuracy. Thus, to find a better balance between computational cost and accuracy, the total bit-FLOPs

selected by the bit selector are regularized, such that:

$$\mathcal{L}_{br} = \frac{1}{N} \sum_{i=1}^{N} max(\frac{B^{\mathcal{Q}}(\boldsymbol{I}_i)}{B_{tar}}, 1), \tag{10}$$

where $B^{\mathcal{Q}}(\cdot)$ denotes the total bit-FLOPs of the quantized network $\mathcal{Q}$ assigned to each synthetic image $\boldsymbol{I}_i$, which is calculated as the weighted number of operations by the weight bit-width and the activation bit-width, $N$ is the mini-batch size, and $B_{tar}$ is the target bit-FLOPs. In this work, we set $B_{tar}$ as the bit-FLOPs of the {5,4}-bit model to compare with existing methods. The overall objective function of our data-free dynamic quantization framework is:

$$\mathcal{L}^{\mathcal{Q}} = \gamma \mathcal{L}_{br} + \frac{1}{N} \sum_{i=1}^{N} \mathcal{L}_{ce}(\mathcal{Q}(\boldsymbol{I}_i), \boldsymbol{y}_i) + \frac{1}{N} \sum_{i=1}^{N} \mathcal{L}_{kd}(\mathcal{P}(\boldsymbol{I}_i), \mathcal{Q}(\boldsymbol{I}_i)), \tag{11}$$

where $\gamma$ is a hyperparameter that balances two loss terms. The final objective is optimized by training the quantized network $\mathcal{Q}$.

**Difficulty-aware mixup** During training, we observe that the easy samples are learned quickly by the dynamic quantization framework, giving minimal contribution to the training afterward. Thus, motivated by Zhang et al. (2017), we adopt mixup to easily fit samples (*i.e.*, samples with lower cross-entropy loss at saturation). We blend two easily-fit images to generate a hard-to-fit image:

$$\tilde{\boldsymbol{I}}_{j_1} = \lambda \boldsymbol{I}_{j_1} + (1 - \lambda) \boldsymbol{I}_{j_2}, \tag{12}$$

$$\tilde{\boldsymbol{y}}_{j_1} = \lambda \boldsymbol{y}_{j_1} + (1 - \lambda) \boldsymbol{y}_{j_2}, \tag{13}$$

where $\lambda \in [0, 1]$ and $(\boldsymbol{I}_{j_1}, \boldsymbol{y}_{j_1})$ and $(\boldsymbol{I}_{j_2}, \boldsymbol{y}_{j_2})$ are pairs randomly drawn from $p\%$ ratio of samples with low $\{\mathcal{L}_{ce}(\mathcal{Q}(\boldsymbol{I}_i), \boldsymbol{y}_i)\}_{i=1}^{N}$ in which ratio $p$ is a hyper-parameter. Followingly, we optimize our network $\mathcal{Q}$ using $\mathcal{L}_{ce}(\mathcal{Q}(\tilde{\boldsymbol{I}}), \tilde{\boldsymbol{y}})$ and $\mathcal{L}_{kd}(\mathcal{P}(\tilde{\boldsymbol{I}}), \mathcal{Q}(\tilde{\boldsymbol{I}}))$ after saturation point.

## 4 EXPERIMENTS

In this section, the proposed framework DynaDFQ is evaluated with various image classification networks to validate its effectiveness. We first describe our experimental settings (Sec. 4.1) and evaluate our framework on CIFAR-10/100 (Sec. 4.2) and ImageNet (ILSVRC12) dataset (Sec. 4.3). Then, we present ablation experiments to analyze each key attribute of our framework (Sec. 4.4).

### 4.1 IMPLEMENTATION DETAILS

**Models** To validate the flexibility of our framework, we perform evaluation with the representative classification models, ResNet-20 (He et al., 2016), ResNet-18 (He et al., 2016), and Mo-bileNetV2 (Sandler et al., 2018) using CIFAR-10/100 (Krizhevsky et al., 2009) and ImageNet (Deng et al., 2009) datasets. DynaDFQ is implemented on top of the baseline DFQ framework, ZeroQ (Cai et al., 2020). For the dynamic bit-width allocation, the bit selector is located after the first two convolutional layers of the network, which are quantized with a fixed bit-width, the highest bit among the $M$ candidates (*i.e.*, $b_M$). In this work, to compare with the fixed 5-bit quantization methods, we set the bit-width candidates near the comparison bit, namely {4,5,6}. Quantization of each bit-width is done using the simple asymmetric uniform quantization function with scale parameters following Jacob et al. (2018); Shoukai et al. (2020); Zhong et al. (2022); Li et al. (2023).

**Training details** All our experiments are implemented using PyTorch Paszke et al. (2019). For data generation, synthetic images are updated for 4000 iterations with batch size of 64 using the Adam Kingma & Ba (2015) optimizer with a learning rate of 0.5 decayed by the rate of 0.1 when the loss stops decreasing for 100 iterations. A total of 5120 images are generated for both CIFAR-10/100 and ImageNet, which is highly lightweight compared to the original dataset (*e.g.*, ∼1M images for ImageNet). For hyperparameters, we set $\beta$=0.1, $K$=2 and $r$=0.5. For the dynamic bit-width allocation, the bit-selector is initialized to output the highest probability for the target bit-width (*e.g.*, 5-bit for ∼5 mixed-precision (MP)). Using the 5120 synthetic images and soft labels, we fine-tune our dynamic quantization framework for 400 epochs with a batch size of 256 for CIFAR-10/100 and 150 epochs, batch size of 16 for ImageNet. The parameters are updated with an SGD optimizer

| Method | G. | Bit-width | Bit-FLOPs (%) | Top-1 (%) |
|---|---|---|---|---|
| Baseline | - | 32 | 100.00 | 94.03 |
| Real Data | - | 5 | 3.52 | 93.96 |
| GDFQ (ECCV'20) | ✓ | 5 | 3.52 | 93.38 |
| Qimera (NeurIPS'21) | ✓ | 5 | 3.52 | 93.46 |
| AdaDFQ (CVPR'23) | ✓ | 5 | 3.52 | 93.81 |
| ZeroQ (CVPR'20) | ✗ | 5 | 3.52 | 91.38[†] |
| DSG (CVPR'21) | ✗ | 5 | 3.52 | 92.73 |
| IntraQ (CVPR'22) | ✗ | 5 | 3.52 | 92.78 |
| HAST (CVPR'23) | ✗ | 5 | 3.52 | 93.43[†] |
| **DynaDFQ (Ours)** | ✗ | ~5 MP | **3.47**±0.06 | **93.87**±0.07 |
| Real Data | - | 4 | 2.62 | 91.52 |
| SQuant (ICLR'22) | - | 4 | 2.64 | 92.24 |
| GDFQ (ECCV'20) | ✓ | 4 | 2.62 | 90.25[‡] |
| Qimera (NeurIPS'21) | ✓ | 4 | 2.62 | 91.26 |
| AdaDFQ (CVPR'23) | ✓ | 4 | 2.62 | 92.31 |
| ZeroQ (CVPR'20) | ✗ | 4 | 2.62 | 84.68[‡] |
| DSG (CVPR'21) | ✗ | 4 | 2.62 | 88.74[‡] |
| GZNQ (CVPRW'21) | ✗ | 4 | 2.62 | 91.30[‡] |
| IntraQ (CVPR'22) | ✗ | 4 | 2.62 | 91.49[‡] |
| HAST (CVPR'23) | ✗ | 4 | 2.62 | 92.36 |
| **DynaDFQ (Ours)** | ✗ | ~4 MP | **2.59**±0.01 | **92.65**±0.05 |

| Method | G. | Bit-width | Bit-FLOPs (%) | Top-1 (%) |
|---|---|---|---|---|
| Baseline | - | 32 | 100.00 | 70.33 |
| Real Data | - | 5 | 3.52 | 70.05 |
| GDFQ (ECCV'20) | ✓ | 5 | 3.52 | 66.12 |
| Qimera (NeurIPS'21) | ✓ | 5 | 3.52 | 69.02 |
| AdaDFQ (CVPR'23) | ✓ | 5 | 3.52 | **69.93** |
| ZeroQ (CVPR'20) | ✗ | 5 | 3.52 | 65.89[†] |
| DSG (CVPR'21) | ✗ | 5 | 3.52 | 67.65 |
| IntraQ (CVPR'22) | ✗ | 5 | 3.52 | 68.06 |
| HAST (CVPR'23) | ✗ | 5 | 3.52 | 69.00[†] |
| **DynaDFQ (Ours)** | ✗ | ~5 MP | **3.44**±0.01 | 69.74±0.03 |
| Real Data | - | 4 | 2.62 | 66.80 |
| SQuant (ICLR'22) | - | 4 | 2.64 | 63.96 |
| GDFQ (ECCV'20) | ✓ | 4 | 2.62 | 63.58[‡] |
| Qimera (NeurIPS'21) | ✓ | 4 | 2.62 | 65.10 |
| AdaDFQ (CVPR'23) | ✓ | 4 | 2.62 | 66.81 |
| ZeroQ (CVPR'20) | ✗ | 4 | 2.62 | 58.42[‡] |
| DSG (CVPR'21) | ✗ | 4 | 2.62 | 62.36[‡] |
| GZNQ (CVPRW'21) | ✗ | 4 | 2.62 | 64.37[‡] |
| IntraQ (CVPR'22) | ✗ | 4 | 2.62 | 64.98[‡] |
| HAST (CVPR'23) | ✗ | 4 | 2.62 | 66.68 |
| **DynaDFQ (Ours)** | ✗ | ~4 MP | **2.57**±0.01 | **67.30**±0.25 |

(a) Results on CIFAR-10      (b) Results on CIFAR-100

[†] reproduced using the official code     [‡] cited from Zhong et al. (2022)

Table 1: **Results of ResNet-20 on CIFAR-10/100.** Bit-FLOPs (%) show the relative bit-FLOPs compared to the FP baseline ResNet20 (42.20G). G. indicates generator architecture-based methods.

with Nesterov using an initial learning rate of $10^{-5}$ decayed by 0.1 every 100 epochs for CIFAR-10/100 and $10^{-6}$ decayed every 30 epochs for ImageNet. The bit selector parameters are updated similarly but with the initial learning rate of $10^{-3}$ and $10^{-4}$ for CIFAR10/100 and ImageNet. To balance different loss terms in Eq. 11, $\gamma$ is set to 100 throughout the experiments. Also, we apply Mixup finetuning from 50% training epochs with a rate $p$ of 25%.

## 4.2 CIFAR-10/100

To evaluate the efficacy of our proposed framework, we compare the results with the existing data-free quantization methods on CIFAR-10/100 with a representative classification network, ResNet-20. Specifically, we compare DynaDFQ with the early DFQ method (ZeroQ (Cai et al., 2020), GDFQ (Shoukai et al., 2020)) and more recent noise optimization-based methods (DSG (Zhang et al., 2021), GZNQ (He et al., 2021), IntraQ (Zhong et al., 2022), HAST (Li et al., 2023)). Also, we compare our framework with generator-based methods (Qimera (Choi et al., 2021), AdaDFQ Qian et al. (2023)). The accuracy of previous arts are obtained from papers or models reproduced with official codes. To make a fair comparison between existing approaches that allocate a fixed universal bit-width and our dynamic mixed bit-width approach, we compare frameworks with respect to the average bit-FLOPs consumed on the test set. As shown in Table 1, DynaDFQ achieves state-of-the-art accuracy on CIFAR-10 even with fewer bit-FLOPs (Table 1a) and achieves the best or second best accuracy on CIFAR-100 with fewer bit-FLOPs (Table 1b). Notably, in several settings, our data-free method outperforms the results of the original data, which demonstrates the superiority of our difficulty-diverse data and also shows that using our data for dynamic quantization allows a better computational cost-accuracy trade-off.

## 4.3 IMAGENET

We evaluate our framework on the further large-scale dataset, ImageNet on widely adopted classification models, ResNet-18 and MobileNetV2. As shown in Table 2a, compared to other DFQ methods on ResNet-18, our framework outperforms the state-of-the-art method by 0.62% on 4-bit with fewer bit-FLOPs. According to Table 2b, on MobileNetV2, DynaDFQ achieves the highest accuracy with a similar amount of bit-FLOPs. Overall, our data-free dynamic framework reduces the accuracy gap with the network trained using the original ImageNet dataset.

| Method | G. | Bit-width | Bit-FLOPs (%) | Top-1 (%) |
|---|---|---|---|---|
| Baseline | - | 32 | 100.00 | 71.47 |
| Real Data | - | 5 | 3.56 | 70.41 |
| GDFQ (ECCV'20) | ✓ | 5 | 3.56 | 66.82‡ |
| Qimera (NeurIPS'21) | ✓ | 5 | 3.56 | 69.29 |
| AdaDFQ (CVPR'23) | ✓ | 5 | 3.56 | 70.29 |
| ZeroQ (CVPR'20) | ✗ | 5 | 3.56 | 69.65‡ |
| DSG (CVPR'21) | ✗ | 5 | 3.56 | 69.53‡ |
| IntraQ (CVPR'22) | ✗ | 5 | 3.56 | 69.94‡ |
| HAST (CVPR'23) | ✗ | 5 | 3.56 | 70.04† |
| **DynaDFQ (Ours)** | ✗ | ∼5 MP | **3.40**±0.08 | **70.41**±0.01 |
| Real Data | - | 4 | 2.54 | 67.89 |
| SQuant (ICLR'22) | - | 4 | 2.58 | 66.14 |
| GDFQ (ECCV'20) | ✓ | 4 | 2.54 | 60.60‡ |
| Qimera (NeurIPS'21) | ✓ | 4 | 2.54 | 63.84 |
| AdaDFQ (CVPR'23) | ✓ | 4 | 2.54 | 66.53 |
| ZeroQ (CVPR'20) | ✗ | 4 | 2.54 | 60.68‡ |
| DSG (CVPR'21) | ✗ | 4 | 2.54 | 60.12‡ |
| GZNQ (CVPRW'21) | ✗ | 4 | 2.54 | 64.50‡ |
| IntraQ (CVPR'22) | ✗ | 4 | 2.54 | 66.47‡ |
| HAST (CVPR'23) | ✗ | 4 | 2.54 | 66.91 |
| **DynaDFQ (Ours)** | ✗ | ∼4 MP | **2.49**±0.01 | **67.47**±0.05 |

(a) Results of ResNet-18

| Method | G. | Bit-width | Bit-FLOPs (%) | Top-1 (%) |
|---|---|---|---|---|
| Baseline | - | 32 | 100.0 | 73.03 |
| Real Data | - | 5 | 12.88 | 72.01 |
| GDFQ (ECCV'20) | ✓ | 5 | 12.88 | 68.14‡ |
| Qimera (NeurIPS'21) | ✓ | 5 | 12.88 | 70.45 |
| AdaDFQ (CVPR'23) | ✓ | 5 | 12.88 | 71.61 |
| ZeroQ (CVPR'20) | ✗ | 5 | 12.88 | 70.88‡ |
| DSG (CVPR'21) | ✗ | 5 | 12.88 | 70.85‡ |
| IntraQ (CVPR'22) | ✗ | 5 | 12.88 | 71.28‡ |
| HAST (CVPR'23) | ✗ | 5 | 12.88 | 71.72 |
| **DynaDFQ (Ours)** | ✗ | ∼5 MP | **12.87**±0.04 | **71.88**±0.05 |
| Real Data | - | 4 | 11.02 | 67.90 |
| SQuant (ICLR'22) | - | 4 | 11.05 | 22.07 |
| GDFQ (ECCV'20) | ✓ | 4 | 11.02 | 51.30‡ |
| Qimera (NeurIPS'21) | ✓ | 4 | 11.02 | 61.62 |
| AdaDFQ (CVPR'23) | ✓ | 4 | 11.02 | 65.41 |
| ZeroQ (CVPR'20) | ✗ | 4 | 11.02 | 59.39‡ |
| DSG (CVPR'21) | ✗ | 4 | 11.02 | 59.04‡ |
| GZNQ (CVPRW'21) | ✗ | 4 | 11.02 | 53.53‡ |
| IntraQ (CVPR'22) | ✗ | 4 | 11.02 | 65.10‡ |
| HAST (CVPR'23) | ✗ | 4 | 11.02 | 65.60 |
| **DynaDFQ (Ours)** | ✗ | ∼4 MP | **11.02**±0.02 | **66.19**±0.46 |

(b) Results of MobileNetV2

Table 2: **Results of ResNet-18/MobileNetV2 on ImageNet.** Bit-FLOPs (%) show the relative bit-FLOPs compared to the FP baseline (1862.54G for ResNet-18 and 314.12G for MobileNetV2).

| Method | Bit-FLOPs (%) | Top-1 (%) |
|---|---|---|
| ZeroQ + DQ | 2.53 | 65.57 |
| IntraQ + DQ | 2.48 | 65.42 |
| DSG + DQ | 2.47 | 65.02 |
| HAST + DQ | 2.47 | 66.88 |
| **DynaDFQ (Ours)** | 2.49 | **67.47** |

(a) Data generation methods on dynamic quantization

| DQ | Difficulty-diverse Generation | Difficulty-aware Mixup | Bit-FLOPs (%) | Top-1 (%) |
|---|---|---|---|---|
|  |  |  | 2.54 | 66.02 |
|  |  | ✓ | 2.54 | 66.05 |
|  | ✓ |  | 2.54 | 66.76 |
|  | ✓ | ✓ | 2.54 | 66.56 |
| ✓ |  |  | 2.48 | 66.39 |
| ✓ |  | ✓ | 2.39 | 66.36 |
| ✓ | ✓ |  | 2.49 | 67.23 |
| ✓ | ✓ | ✓ | **2.49** | **67.47** |

(b) Ablation study on each attribute

Table 3: **Ablation study of DynaDFQ** done on ImageNet with ResNet-18 (∼ 4MP). We also compare different data generation methods on dynamic quantization (DQ) using our dynamic network.

## 4.4 ABLATION STUDY

**Effect of our data generation** We first present the effectiveness of our sample generation scheme for dynamic quantization by comparing it with the existing data generation techniques for data-free quantization. Table 3a shows that the existing data generation schemes for DFQ fail to effectively allocate bit-widths compared to ours. The results indicate that our synthetic data successfully simulates and controls realistic/plausible difficulty of different levels, and thus enables dynamic quantization to effectively assign computational costs for unseen real input images without real data.

To further verify the effectiveness of our generated samples, we conduct an ablation study on our difficulty-diverse generation scheme. As shown in Table 3b, while our generated data are also helpful for fixed quantization (+0.74%), the benefits are maximally excavated with a dynamic quantization framework (+1.21%) with fewer bit-FLOPs. The results indicate that it is important to generate diverse samples with respect to difficulty, especially for dynamic quantization.

**Effect of our dynamic quantization framework** We validate the effectiveness of our dynamic quantization framework (DQ). As in Table 3b, dynamic quantization alone provides a slightly better trade-off with the baseline data (+0.37%). However, when we train the dynamic quantization framework with our synthetic data, the accuracy increases by a large margin (+1.21%). Moreover, using mixup technique during the dynamic quantization fine-tuning process gives additional gain (+0.24%). The results imply that our difficulty-diverse data generation, along with our dynamic framework, is effective in terms of the trade-off between bit-FLOPs and test accuracy, corroborating the observation in Figure 1 of images of different difficulty exhibiting different trade-off.

**Effect of the hyper-parameters** The results of Table 4 justify our selection of hyper-parameters: the number of similar classes $K$ and diversifying ratio $r$ for the data generation process and $p$ for mixup fine-tuning. We find that the diversifying ratio of 0.50 with two similar classes produces the

| $K$ | | 2 | | | 3 | | 10 |
|---|---|---|---|---|---|---|---|
| $r$ | 0.25 | 0.50 | 1.00 | 0.25 | 0.50 | 1.00 | 0.50 |
| Bit-FLOPs (%) | 2.47 | 2.49 | 2.45 | 2.50 | 2.50 | 2.47 | 2.54 |
| Top-1 (%) | 67.16 | **67.47** | 66.50 | 67.32 | 67.45 | 65.02 | 67.01 |

(a) $K$ and $r$

| $p$ | 25 | 50 | 75 | 100 |
|---|---|---|---|---|
| Bit-FLOPs (%) | 2.49 | 2.48 | 2.48 | 2.51 |
| Top-1 (%) | **67.47** | 67.27 | 67.05 | 67.25 |

(b) $p$

Table 4: **Effect of the hyper-parameters** on the top-1 accuracy and bit-FLOPs of 4-bit ResNet-18 on ImageNet. $K$ and $r$ are used in data generation process and $p$ is used for fine-tuning process.

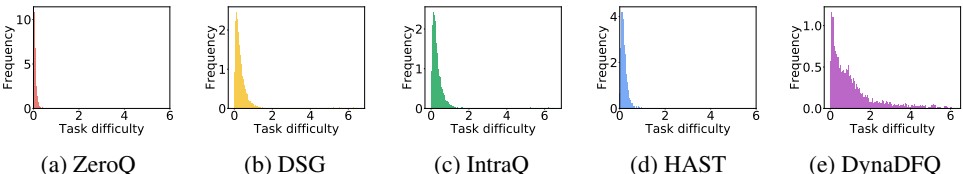

(a) ZeroQ      (b) DSG      (c) IntraQ      (d) HAST      (e) DynaDFQ

Figure 3: **Classification difficulty distribution of generated samples** for ImageNet using Mo-bileNetV2. Our approach generates more *diverse* synthetic data in terms of classification difficulty, which is measured by the entropy of output prediction of each sample.

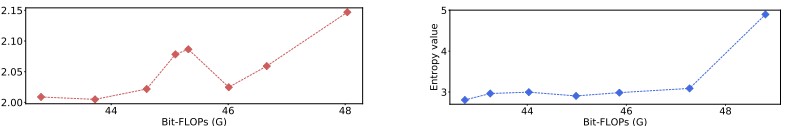

Figure 4: **Bit-width allocation for images of different difficulty.** For a given sample of (left) test images and (right) generated train images, assigned bit-FLOPs and output entropy are plotted. Samples with higher entropy tend to be assigned with overall higher bit-FLOPs.

best results, while a too-large $r$ or $K$ results in an accuracy drop. This is because a large $r$ and $K$ will lead to a set of generated images that mostly consist of difficult images. It is important for the dynamic network to see images of diverse difficulties (both easy and difficult) during training. Also, we find that applying mixup to 25% of images with low difficulty outputs the best result.

### 4.5 QUALITATIVE RESULTS

We compare the difficulty diversity of different data-generation methods in Figure 3. Compared to existing methods, our approach produces more diverse data in terms of difficulty, which hints that the diversity in difficulty serves as a key factor for data-free dynamic quantization. To better demonstrate the efficacy of our dynamic quantization and plausibly difficult sample generation, we visualize the bit allocation results of different difficulty images in Figure 4. One can observe that the overall samples with high (low) entropy tend to be assigned with more (less) bit-FLOPs. Also, the results show that our synthetic images can fairly model the difficulty of real test images.

## 5 CONCLUSION

In this paper, we present the first dynamic data-free quantization method that allocates different bit-widths to different images without any access to the original training data. Despite the considerable merits, dynamic quantization remains unexplored within data-free scenarios. One reason is that, while dynamic quantization requires training with images of different task difficulty levels for effective sample-specific allocation of computational costs, existing data-free approaches fail to synthesize difficulty-diverse data. To this end, we aim to generate difficulty-diverse set of images by assigning random soft labels to each image. However, the randomly sampled soft label is not reliable, as the semantic relation between classes is neglected. Therefore, we assign realistic soft labels considering the class similarity using classification layer weights to ensure the plausibility of images. Experimental results demonstrate the effectiveness of our scheme; we achieve consistent accuracy gain with fewer computational resources.

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

## A  DIFFICULTY DIVERSE GENERATION ANALYSIS

The diversity in difficulty level of images, measured with prediction entropy, can be controlled by the number of similar classes $K$ and the diversifying ratio $r$. The results in Figure 5 show varying entropy distributions by our hyperparameter selection. According to the figure, using a large $r$ or large $K$ produces a set of images that relatively lack easy images. This can cause the dynamic network to inefficiently allocate computational resources, as shown in the main manuscript. Thus, it is important to collect both easy and difficult images to train the dynamic quantization framework, which justifies our choice of $r = 0.5$ and $K = 2$.

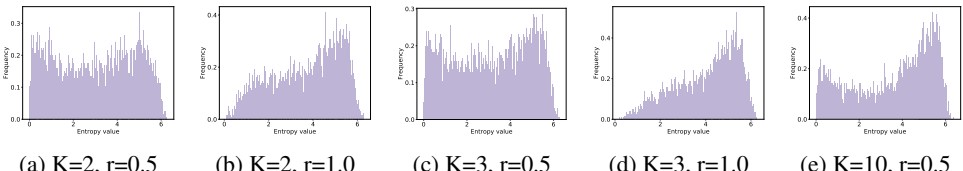

(a) K=2, r=0.5    (b) K=2, r=1.0    (c) K=3, r=0.5    (d) K=3, r=1.0    (e) K=10, r=0.5

Figure 5: **Classification difficulty distribution of synthetic data** for ImageNet using ResNet-18.

## B  BIT-WIDTH CANDIDATE ANALYSIS

We investigate the impact of using different combination of candidate bit-widths, as shown in Table 5. The results demonstrate that different combinations of candidates can provide flexibility in trade-offs between computational cost and accuracy. Compared to using three candidates ($M = 3$) $\{3, 4, 5\}$, adding candidates to ($M = 5$) for the bit selector results in an accuracy loss, but the bit-FLOPs are largely reduced. However, reducing the number of candidates ($M = 2$) leads to an accuracy drop with a small bit-flops reduction, which indicates that two candidates are insufficient to merit from the dynamic framework.

| Bit-width candidates | Bit-FLOPs (%) | Top-1 (%) |
|:---:|:---:|:---:|
| $\{3, 4, 5\}$ | 2.49 | 67.47 |
| $\{3, 5\}$ | 2.48 | 66.85 |
| $\{2, 3, 4, 5, 6\}$ | 2.41 | 65.73 |

Table 5: **Ablation study of DynaDFQ** of $\sim$4 MP on ImageNet with ResNet-18.

## C  ANALYSIS ON MIXUP

We additionally analyze the impact of the mixup technique to our framework in Figure 6. The figure shows the curves of the loss terms during training. After 50% of epochs, when the mixup is applied, we compare the loss using the mixed-up images (blue line) and the loss value of the images before the mixup (orange line). The results show that mixup reduces the loss terms $\mathcal{L}_{ce}^{Q}$ (Figure 6a) and $\mathcal{L}_{kd}$ (Figure 6b). Notably, as shown in Figure 6c, bit regularization loss increases after mixup is applied and remains large for mixed-up images. The reason is that, since we apply mixup on easy samples to further promote the difficulty diversity the network is exposed to, the overall difficulty of the synthetic training set is enhanced. Thus, it is natural to leverage higher bit-widths for difficult samples, and therefore, the bit regularization loss term increases.

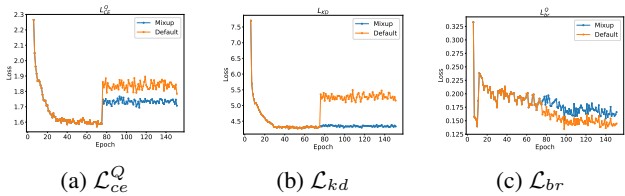

(a) $\mathcal{L}_{ce}^{Q}$    (b) $\mathcal{L}_{kd}$    (c) $\mathcal{L}_{br}$

Figure 6: **Learning curves of losses using mixup** for ImageNet using ResNet-18.

# D  ADDITIONAL VISUALIZATION

We provide additional visualization results to validate the generalizability of our observation of the main manuscript; difficult images are confusing between similar classes, and the class of the classification layer can function as a prototype of that class. According to Figure 7, which displays more examples of difficult and easy images in the original dataset (ImageNet), difficult images tend to be confusing among semantically similar classes. We visualize the two images with the highest and lowest prediction entropy of each class, respectively. Moreover, we additionally validate the correlation between the similarity of class weights and the actual co-occurrences of classes in prediction output, as visualized in Figure 8. The results demonstrate that we can estimate the semantic similarity between classes with weights of the classification layer without accessing the original data.

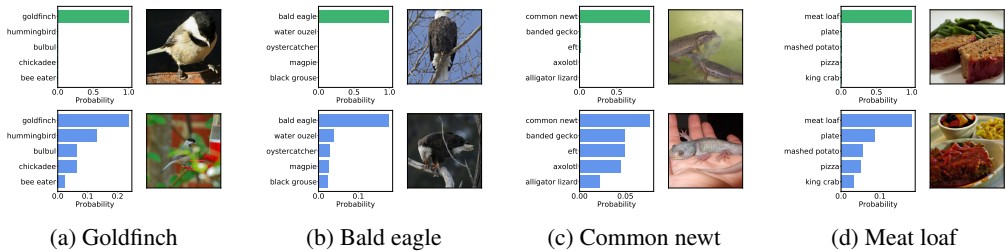

(a) Goldfinch (b) Bald eagle (c) Common newt (d) Meat loaf

Figure 7: **More examples of easy and difficult images in ImageNet.**

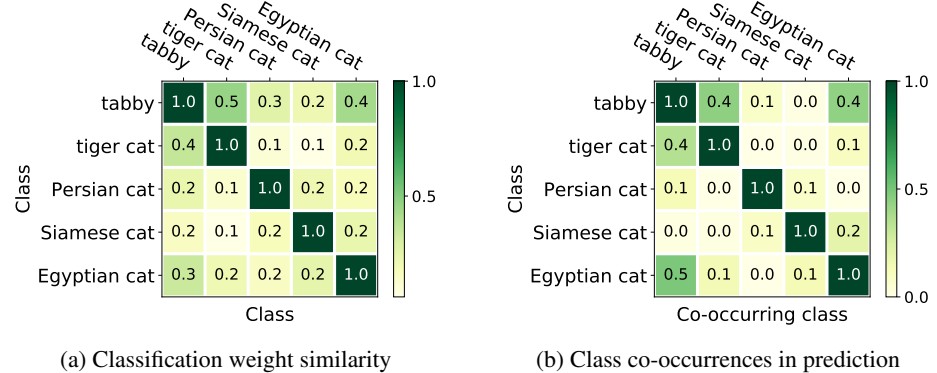

(a) Classification weight similarity (b) Class co-occurrences in prediction

Figure 8: **Additional validation on the correlation between class weight similarity and class co-occurrences in prediction** on five cat classes of ImageNet.

# E QUALITATIVE RESULTS

In Figure 9, our synthesized images using the baseline ResNet-18 (ImageNet) are presented. While samples generated with $K = 1$ exhibit low prediction entropy, samples generated with $K = 2$ are relatively confusing for the classification network. Notably, samples with $K = 1$ are high-fidelity images with details while samples with $K = 2$ tend to have indistinct structures of a certain class, making it confusing between similar groups of classes. The results show that our method generates a diversified set of plausible images with varying levels of difficulty.

Furthermore, we analyze the impact of $K$ on the generated images. In Figure 10, the synthesized image of the same top-1 class becomes more complex for larger $K$. This indicates that a large $K$ can produce an overly confusing image due to the assigned irrelevant classes, such as *window screen* class to generate a confusing *dining table* image. $K = 2$ is sufficient to generate an image that is confusing for the classification network.

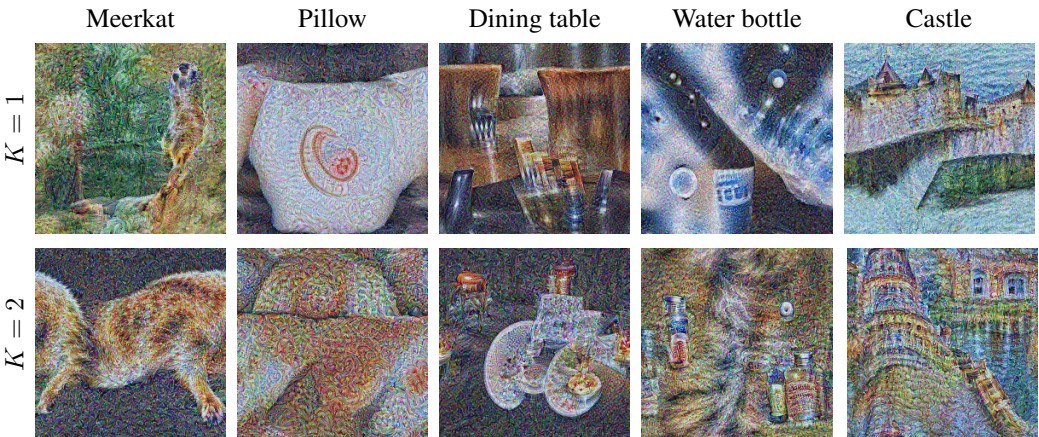

Figure 9: **Generated samples of DynaDFQ for ImageNet with ResNet-18.** The top row consists of samples generated with $K = 1$, and bottom row of samples with $K = 2$. Images of each column have the same top-1 class.

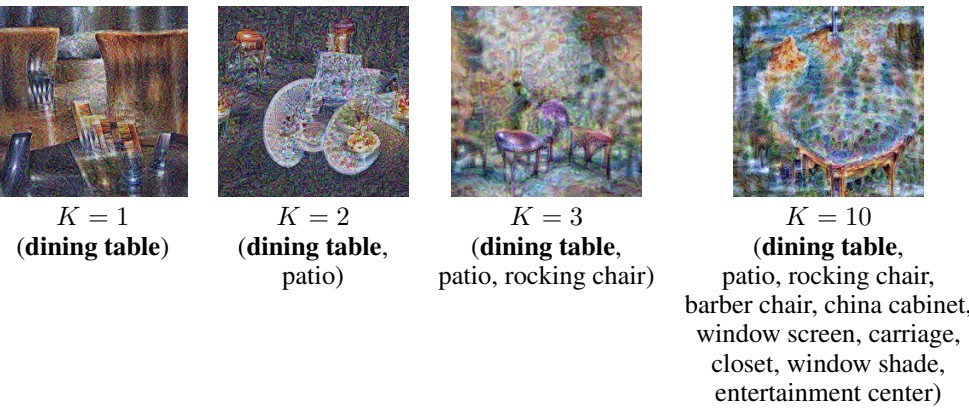

Figure 10: **Generated samples of the same top-1 class (dining table) using different Ks.** Images are synthesized from ImageNet-pretrained ResNet-18. Classes in brackets denote the assigned top-K classes.

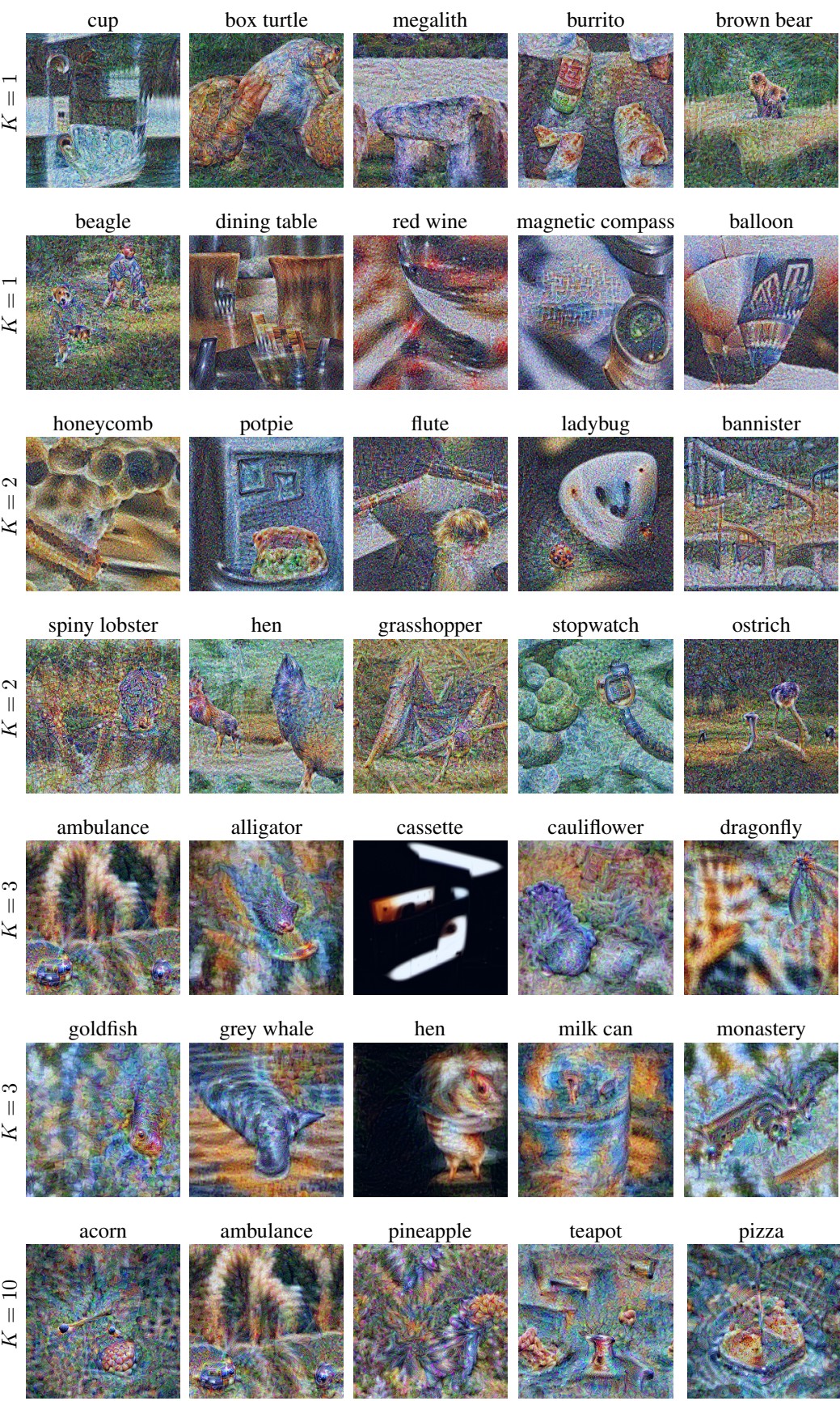

