# OpenReview forum: "Diversity, Plausibility, and Difficulty: Dynamic Data-Free Quantization"
_ICLR.cc/2024/Conference — ICLR 2024 Conference Withdrawn Submission_

### Official Review · Reviewer_Esi8 · 2023-10-29

**Soundness:** 2 fair
**Presentation:** 3 good
**Contribution:** 2 fair
**Rating:** 5
**Confidence:** 5

**Summary:**

This paper focuses on two different subjects, DFQ (Data-Free Quantization) and DQ (Dynamic Quantization). DFQ is a research topic that quantizes neural networks without any data because the training dataset can be hard to access in real-world scenarios. DQ is finding optimal bit setting while inference, because the difficulty of each image is different, and a trade-off between computational cost and accuracy is also different accordingly. The paper proposes a method called DynaDFQ, which can solve the DFQ problem under DQ settings.

The paper adopts a bit selector which finds out the optimal bit setting of a given input image. In the DQ problem, the bit selector should be trained with real images, because it should be able to classify whether the given input is easy enough to infer with a lower bit. Thus, there is an obstacle to train Bit selector under DFQ scenarios, because the training dataset is not available. To solve this problem, the paper tries to generate images that show various difficulties. The paper assumes that a difficult image refers to a confusing image that is difficult to distinguish into one of several similar classes. Therefore, the paper adopts soft labels to generate difficult images. By doing so, generated images with soft labels can be used to train a bit selector. In addition, they also help enhance the performance of quantized networks.

**Strengths:**

- The paper is well-written and easy to follow.
- The claims of the paper are persuasive and it is described well with several formulas and figures.
- The paper adopts dynamic quantization to data-free quantization for the first time and reported results show better performance than other previous works in both accuracy and bit-flops.

**Weaknesses:**

- In Formula 5, the paper uses dot product operation for comparing the similarity between two arbitrary classes. However, because dot product operation produces results that are not normalized, there can be some noise in similarity results.
- The concept of soft labels that are used for promoting the difficulty of images is already used in other works (e.g., superposed embedding in Qimera). However, the paper doesn’t explain what’s different between previous works and the proposed ones.
- The paper measures the cost of each network with bit-flops. However, because DynaDFQ has an additional operation, which is called bit selector, further realistic analysis about costs should have been provided.

**Questions:**

- In terms of inference time, especially with low-bit and lightweight neural networks, it is important to reduce the number of accesses to DRAM. However, the bit selector needs other weight parameters that aren’t related to the original network’s weight parameters. Also, the output of the bit selector isn’t reused in the original network. Furthermore, while operating the bit-selector, the hardware isn’t likely to be fully utilized. These problems may lead to harm to the pipelining of the inference. Can the authors’ opinions be provided?
- DQ is for selecting whether to use a lower bit or not, according to the difficulty of each image. That is, DQ is finding better accuracy-efficiency trade-offs. However, in Table 3 (a), accuracies of ZeroQ, IntraQ, and DSQ are improved when DQ is applied. Can the authors’ explanation be provided?
- HAST tried to generate hard samples too, which is different from the method proposed in this paper. When entropy is used as a measure of difficulty, the difficulty of images generated with DynaDFQ is higher than that of HAST. Can the authors measure the difficulties of images with GHM, which is used in HAST? It will be helpful to find out which method is more suit for measuring the difficulties of images.
- It seems that DynaDFQ is only for processing one image at once. If a batch of inputs is delivered, how does the bit selector work?
- The performance of the bit selector itself isn’t described well. Can the authors provide a performance of the proposed method without a bit selector, to understand whether a bit selector works well or not? Also, can the precision and the recall of a bit selector be provided?
- The paper builds soft labels only with classes that show high similarity. How does performance change if soft labels are built with arbitrary classes, or classes which consist of both high similarity and low similarity?

---

### Official Review · Reviewer_P8Hw · 2023-10-30

**Soundness:** 3 good
**Presentation:** 3 good
**Contribution:** 2 fair
**Rating:** 3
**Confidence:** 5

**Summary:**

This paper proposes the first Data-Free Dynamic Quantization framework (DynaDFQ) that learns to allocate bit-widths dynamically to the input image according to its difficulty, without the need of the original training data. And this work claims that two aspects of difficulty should be considered: diversity and plausibility. To formulate both aspects in the image synthesis, the authors make use of readily available information that has not been exploited by previous works: class similarity.

**Strengths:**

This paper proposes a relatively complete dynamic mixed-precision data-free quantization framework by synthesizing data, with good presentation and reasonable experimental settings.

**Weaknesses:**

1. The contribution of this paper is very limited. In my opinion, it is more like a simple patchwork of several methods. For instance, diversity based on image synthesis has been used in many literatures, e.g., IntraQ [a] and DSG [b]. Dynamically allocating bit-widths is no improvement over the current methods. As for soft-label and mixup [c], it is a very common trick.
2. Even though many tricks are used, the experimental results are not much improved compared to SOTAs. There is no more fine-grained comparison between diversity and plausibility in the ablation experiment.
3. There are few related works after 2021, and some important works are not mentioned, e.g., [d] and [e].

[a] Intraq: Learning synthetic images with intra-class heterogeneity for zero-shot network quantization.
[b] Diversifying sample generation for accurate data-free quantization
[c] mixup: Beyond empirical risk minimization
[d] Unified Data-Free Compression: Pruning and Quantization without Fine-Tuning
[e] Spiq: Data-free per-channel static input quantization

**Questions:**

1. How is the dynamic quantization part used in training? Specifically, Eq.8 does not appear in any subsequent loss functions. So how will the bit-width be trained based on Eq.11?
2. It is recommended to add a training algorithm with pseudo-code to better understand the work process.

---

### Official Review · Reviewer_Lwg9 · 2023-11-01

**Soundness:** 2 fair
**Presentation:** 3 good
**Contribution:** 2 fair
**Rating:** 3
**Confidence:** 5

**Summary:**

This paper combines the data-free quantization and dynamic quantization.

**Strengths:**

1, Not mistake in motivations.
2, Further improving the performance of ZSQ.

**Weaknesses:**

Some arguments in this paper should be corrected. For example:
1, The arguments of “we find that the quantization sensitivity is related to the classification difficulty of samples.” are more likely a consensus. The author should find some reference to support their findings instead of claiming this finding is their own.

2, The argument of “To formulate both aspects in our image synthesis, we make use of readily available information that has not been exploited by previous works: class similarity” is more likely an overstatement. It should be that the class similarity has not been exploited by previous ZSQ methods. In fact, the authors themselves say this has been proposed by Saito et al., 2019.

The novelty is marginal:


This paper simply combines two parts: ZSQ and dynamic quantization.

I cannot find any connection between the proposed ZSQ method and dynamic quantization. In my view, the ZSQ itself also works even without dynamic quantization. However, I can’t find any experimental results about this. Moreover, the part of dynamic quantization is more likely an established method. Currently, considering ICLR is a high-level conference, the novelty is not enough.


I do understand the main idea of this paper is to use dynamic quantization to prompt performance. However, the use of dynamic quantization typically brings improvements. It should show some results without using dynamic quantization to demonstrate the effectiveness of the ZSQ part of this paper.

**Questions:**

Questions:

1, Some related works are missing. For example, it seems that [1] also uses mixup to augment the data. The author should cite this paper. The comparison would be too strict as this paper is more like a contemporary work. However, it would be better if the author cited this paper for completeness.

2, The author should provide some results without using dynamic quantization, which is very important for a fair comparison.

[1] TexQ: Zero-shot Network Quantization with Texture Feature Distribution Calibration

---

### Official Review · Reviewer_TLwm · 2023-11-01

**Soundness:** 4 excellent
**Presentation:** 3 good
**Contribution:** 2 fair
**Rating:** 5
**Confidence:** 5

**Summary:**

This paper presents the first dynamic data-free quantization method that allocate bit-widths dynamically to the input image according to its difficulty, without any access to the original training data. Based on the observation that the similarity of class weights from the classification layer can be used to measure the similarity between classes of the original data, this paper gets some hard samples with soft labels, where the probabilities are allocated to a group of similar classes, and some easy samples with one-hot labels. After that, the paper use a lightweight bit selector for bit-width allocation and use the mixup technique for easy samples during training.

**Strengths:**

1. The first work to combine dynamic quantization and data-free quantization.

2. Based of the observation that class similarity information can be extracted from the classification layer of the original pre-trained network, this paper gets hard samples with soft labels, where the probabilities are allocated to a group of similar classes.

**Weaknesses:**

1, the authors do not prove that the bit-width allocated by the bit selector matches the true quantization sensitivity. If I am wrong, please let me know.

2. the mixup technique in the paper has been proposed in [1].

3. there is only a comparison between different DFQ methods+DQ with DynaDFQ in Table 3(a) in the paper, not a comparison of other DFQ methods with DynaDFQ at fixed bits (e.g. a4w4).  I want to see some results if dynamic quantization is not used.

4. hard samples are obtained based on class similarity in the paper, but there are previous DFQ methods on hard sample generation (e.g. HAST [2]). This paper lacks comparative experiments using hard samples generated in other DFQ methods for training.

3 and 4 mean to find that the effect of the hard sample method in this paper and the effect of dynamic method in this paper, respectively.

5. Is entropy in section 3.2 useful in the process of sample synthesis or model training? Why put it in the section difficulty diversity?

[1] MixMix: All You Need for Data-Free Compression Are Feature and Data Mixing, https://arxiv.org/abs/2011.09899 (ICCV’21)
[2] Hard Sample Matters a Lot in Zero-Shot Quantization, https://arxiv.org/abs/2303.13826 (CVPR’23)

**Questions:**

See weaknesses

**Details Of Ethics Concerns:**

The paper is not innovative enough and experiments are not sufficient.